# Cenosphere-Based Zeolite Precursors of Lutetium Encapsulated Aluminosilicate Microspheres for Application in Brachytherapy

**DOI:** 10.3390/ma15197025

**Published:** 2022-10-10

**Authors:** Tatiana Vereshchagina, Ekaterina Kutikhina, Sergei Vereshchagin, Olga Buyko, Alexander Anshits

**Affiliations:** 1Federal Research Center “Krasnoyarsk Science Center of Siberian Branch of the Russian Academy of Sciences”, Institute of Chemistry and Chemical Technology, 50/24 Akademgorodok, Krasnoyarsk 660036, Russia; kutikhina@icct.ru (E.K.); snv@icct.ru (S.V.); bujko86@gmail.com (O.B.); anshits@icct.ru (A.A.); 2Department of Chemistry, Siberian Federal University, Svobodny Av. 79, Krasnoyarsk 660041, Russia; 3Research Department, Siberian Federal University, Svobodny Av. 79, Krasnoyarsk 660041, Russia

**Keywords:** cenospheres, zeolites, brachytherapy, lutetium aluminosilicate microspheres

## Abstract

Coal fly ash hollow aluminosilicate microspheres (cenospheres) of stabilized composition (glass phase—95.4; (SiO_2_/Al_2_O_3_)_glass_—3.1; (Si/Al)_at._ = 2.6) were used to fabricate lutetium-176 encapsulated aluminosilicate microspheres as precursors of radiolabeled microspheres applied for selective irradiation of tumors. To incorporate Lu^3+^ ions into cenosphere’s aluminosilicate material, the following strategy was realized: (i) chemical modification of cenosphere globules by conversion of aluminosilicate glass into zeolites preserving a spherical form of cenospheres; (ii) loading of zeolitized microspheres with Lu^3+^ by means of ion exchange 3Na^+^ ↔ Lu^3+^; (iii) Lu^3+^ encapsulation in an aluminosilicate matrix by solid-phase transformation of the Lu^3+^ loaded microspheres under thermal treatment at 1273–1473 K. Two types of zeolitized products, such as NaX (FAU) and NaP1 (GIS) bearing microspheres having the specific surface area of 204 and 33 m^2^/g, accordingly, were prepared and their Lu^3+^ sorption abilities were studied. As revealed, the Lu^3+^ sorption capacities of the zeolitized products are about 130 and 70 mg/g Lu^3+^ for NaX and NaP1 microspheres, respectively. It was found that the long-time heating of the Lu^3+^-loaded zeolite precursors at 1273 K in a fixed bed resulted in the crystallization of monoclinic Lu_2_Si_2_O_7_ in both zeolite systems, which is a major component of crystalline constituents of the calcined microspheres. The fast heating–cooling cycle at 1473 K in a moving bed resulted in the amorphization of zeolite components in both precursors and softening glass crystalline matter of the NaX-bearing precursor with preserving its spherical form and partial elimination of surface open pores. The NaX-bearing microspheres, compared to NaP1-based precursor, are characterized by uneven Lu distribution over the zeolite-derived layer. The precursor based on gismondin-type zeolite provides a near-uniform Lu distribution and acceptable Lu content (up to 15 mol.% Lu_2_O_3_) in the solid phase.

## 1. Introduction

Microspheres with encapsulated beta-emitting radioisotopes (Y-90, Lu-177, Ho-166, Dy-164, etc.), are used for selective internal radiation therapy of tumors (SIRT, brachytherapy) by means of radioembolization in which the radiolabeled microspheres are injected into the artery that leads to the tumor [1,2]. The Y-90 (half-life = 64.1 h) bearing microspheres are most commonly applied for selective irradiation of liver metastases [3]. One of the commercially available products is TheraSpheres™ (Nordion, Ottawa, ON, Canada) consisting of yttrium aluminosilicate glass microspheres [2]. Among radioisotopes, Lu-177 (half-life = 6.7 d) is considered to be also the promising beta-emitter for application in brachytherapy [4]. Usually, Lu-177 is used together with Y-90 as a radioisotope facilitating diagnostic because it is capable to emit a detectable and non-hazardous gamma irradiation [5].

As reported in previous studies, the microspheres themselves can be made of glass, ceramic, ion exchange resins, biodegradable polymers, etc. [2]. The method of incorporation of radionuclides in the microsphere carrier depends on the properties of the microsphere material. As a rule, the microspheres comprising of organic matter bind a radioisotope at expense of the ion exchange, chelating with functional groups (-OH, -NH_2_, -SH, -COOH) and other interactions which do not provide the reliable immobilization of the trapped radionuclide [2]. Conversely, in the case of glass microspheres, the stable precursors of the beta-emitter (e.g., Y-89, Lu-176, P-31) are incorporated in the microsphere material at the stage of glass preparation followed by neutron activation to form the beta-emitters Y-90, Lu-177, P-32 [1,6]. The latter approach ensures the stronger binding of the beta-emitter in the glass matrix compared to the polymer-based method reducing the risk of the beta-emitter release into the organism [7]. Microsphere yttrium aluminosilicate glasses with a composition of 17 mol.% Y_2_O_3_—19 mol.% Al_2_O_3_—64 mol.% SiO_2_ were fabricated by the melting-quenching process followed by grinding a glass monolith and spheroidization of glass particles in plasma [6]. Chemically durable ceramic Y_2_O_3_ and YPO_4_ microspheres with a higher yttrium content are produced by Kawashita et al. [8] using a high-frequency induction thermal plasma melting technique. Thus, all the synthetic routes to prepare the durable radiolabeled microspheres exploits the high-temperature expensive procedures.

An alternate resource- and energy-saving approach to fabricate microspheres with encapsulated lanthanides can be implemented by using hollow aluminosilicate glass-crystalline microspheres (cenospheres) generated at coal-firing power plants [9,10,11]. This consideration is supported by the experimental verification of producing (i) cenosphere fractions of stabilized chemical, phase and granulometric composition with predictable properties by fine classification of cenosphere concentrates [12] and (ii) functional materials for various high-tech applications [13,14,15]. The cenospheres are comprised of aluminosilicate glass (up to 95 wt.%) with micro- and submicrosized inclusions of crystal phases, which can be chemically modified resulting in porous microspheres, composite sorbents, mineral-like compounds (zeolites, feldspars, feldspathoids, etc.) [16]. The significant prerequisite to recommending cenospheres as a matrix for the Lu (III) encapsulation is based on the previous experience of using cenosphere-based sorbents for immobilizing the water-soluble forms of ^137^Cs and ^90^Sr radionuclides [17,18] and lanthanides [19] in silicate/aluminosilicates phases by thermal treatment of metal loaded sorbents. By doing this, the sorbed metal cations are incorporated into crystal phases via interaction with Si and/or Al of the sorbent material. The calcined material has a low leachability/solubility of the sorbed metal in the aqueous medium, and this parameter is very important from the point of view of satisfying the requirements for beta-emitting microspheres [4].

Thus, the cenosphere-based methodology to incorporate Lu(III) into the cenosphere’s aluminosilicate glass with preserving its spherical form can include the following steps: (i)Activation of glass surfaces (internal and external) by elaboration of porosity by applying an acid or alkaline treatment;(ii)Creation of specific binding sites for metal cations, such as nucleophilic or chelating functional groups, and the ability for the ion-exchange interaction;(iii)Lu(III) sorption or ion exchange;(iv)Solid-phase transformation of Lu(III) loaded microspheres under heating at a given temperature.

Under applied conditions, Lu(III) is expected to enter in micro-/nano-sized crystal phases encapsulated in glass. 

In the frame of the experimental approach mentioned above, the conversion of the cenosphere’s aluminosilicate glass into zeolite structures (zeolitization) seems to be the promising method for Lu(III) immobilization in an aluminosilicate matrix. Certain regularities of the zeolitization process resulting in hollow-structured monozeolite products with preserving a spherical form of the cenosphere-based template have been established [20,21]. The SiO_2_/Al_2_O_3_ ratio in aluminosilicate glass, temperature, alkaline molarity, solid-to-liquid ratio, and agitation mode of the reaction mixture are the key factors for zeolite crystallization of a given structural type in the NaOH-H_2_O-(SiO_2_-Al_2_O_3_)_glass_ system, where (SiO_2_-Al_2_O_3_)_glass_ is cenosphere’s glass. This provided a basis for the direct transformation of a sphere-shaped glassy material into hollow microspheres with composite glass/zeolite shells based on a nearly monozeolite phase. At present, progress has been achieved in the prevailing formation of low-silica zeolites, such as NaP1 (GIS) [20], NaX (FAU) [21], and analcime (ANA) [22]. It is obvious that the crystallization of pure zeolite phases will create a uniform zeolite covering with crystals of controlled topology and, respectively, morphology. Ultimately, this will promote the homogeneous distribution of metal cations, which are to be fixed in the bulk of the aluminosilicate matrix by means of the “ion exchange−high-temperature recrystallization” procedure. The phase transformation of Na, Lu-zeolite at high temperatures is an important stage for Lu incorporation in its own crystal phase providing low Lu solubility in a blood-like liquid. Another significant parameter of the zeolite precursor, which affects the content of the incorporated target ion, is its ion exchange capacity. 

Taking into account the requirements of the zeolite precursor, such as (i) the homogeneity of metal distribution over the microsphere shell and (ii) the sufficient metal loading comparable to the same one of the reported microspheres, for example, yttrium-aluminosilicate microspheres (17–22 mol.% Y_2_O_3_) [6], NaX and NaP1 zeolites were selected to be synthesized in the cenosphere-based NaOH-H_2_O-(SiO_2_-Al_2_O_3_)_glass_ system. The distinctive features of the zeolites are as follows [23]:(1)Zeolite NaX is a molecular sieve with faujasite topology, three-dimensional pore structure with 7.4 Å diameter pores, and Si/Al = 1–1.5; due to the large pore size diffusion of lanthanide aqua complexes in the framework channels is not hindered facilitating the Ln^3+^ exchange up to 85% level at room temperature [24]; the irreversible rare earth ion exchange is characteristic for synthetic faujasites [25];(2)Zeolite NaP1 is a molecular sieve with gismondin topology, three-dimensional pore structure with 3.1 Å × 4.4 Å and 2.8 Å × 4.9 Å pores, and Si/Al = 1.1–2.5; the 3Na^+^—La^3+^ exchange on fly ash derived NaP1 zeolite with the sorption capacity of 58 mg/g is reported [26,27].

Thus, the paper is aimed at the preparation of lutetium-encapsulated aluminosilicate microspheres with homogeneous Lu distribution and high Lu loading by implementing the three-step process, such as (i) formation of zeolite coverings on the surface of cenosphere globules, (ii) Lu^3+^ entering into zeolite structure by the ion exchange 3Na^+^ ↔ Lu^3+^, and (iii) solid-phase transformation/recrystallization of the lutetium loaded zeolites under heating. The stable isotope Lu-176 is used as a stable precursor of β-emitting Lu-177.

## 2. Materials and Methods

### 2.1. Chemicals and Materials

Sodium hydroxide, NaOH (“p.a.”, Vekton, Russia) and lutetium (III) nitrate, hexahydrate, Lu(NO_3_)_3_⋅6H_2_O (“puriss.”, Chemcraft, Russia) were used in zeolite syntheses and preparation of Lu^3+^ bearing solutions, accordingly.

The initial cenosphere fraction including hollow globules of −180 + 80 μm in size (further marked as (SiO_2_-Al_2_O_3_)_glass_) was the product of separation of a coal fly ash cenosphere concentrate resulting from combustion of Kuznetsk coal (Russia) according to the reported procedures [12]. Chemical and phase compositions (wt.%) of the cenospheres were as follows: SiO_2_—67.6, Al_2_O_3—_21.0, Fe_2_O_3—_3.2, CaO+MgO+Na_2_O+K_2_O—7.7; quartz—3.4, mullite—0.8, calcite—0.5, glass phase—95.4; (SiO_2_/Al_2_O_3_)_glass—_3.1, (Si/Al)_at._ = 2.6; SSA = 0.2 m^2^/g. SEM images of the cenosphere globules and a PXRD pattern of the cenosphere material are given in Figure 1.

### 2.2. Hydrothermal Synthesis of Zeolite Layers (Zeolitization)

The reaction mixtures with a liquid-to-solid (L/S) ratio of about 7/1 (*v*/*v*) were prepared by adding the cenospheres (~5.0 g) to 1.5–2.5 mol/L NaOH resulting in two NaOH-H_2_O-(SiO_2_-Al_2_O_3_)_glass_ systems of different molar compositions (Table 1). The reaction mixtures were transferred into a Teflon-lined stainless-steel autoclave (“Beluga”, Premex AG, Switzerland) which was tightly closed without preliminary evacuation. The syntheses were carried out at 353–393 K and autogenous pressure for 24–48 h without stirring (No. 1 in Table 1) and with agitation at a rate of 50 rpm (No. 2 in Table 1). Then the solid products were separated by filtration, washed with distilled water several times until neutral reaction of a supernatant occurred, and isolated by decantation. The sediments were dried at 383 K in air and products thus prepared are denoted as NaX and NaP1 (Table 1).

### 2.3. Preparation of Lu(III) Loaded Microspheres

The Lu(III) loading of the zeolitized microspheres (Table 1) were carried out via the ion exchange 3Na^+^ ↔ Lu^3+^ upon contacting the specimen (0.0500 ± 0.0005 g) with Lu(NO_3_)_3_ solution of 100–1000 mg/L Lu^3+^ at agitation (V = 40 mL; pH~6.0; ambient temperature; τ = 24 h). The equilibrium Lu^3+^ quantity in the solid phase (Q_e_, mg/g) was determined as Q_e_ = (C_o_ − C_e_)∙V/m, where C_o_ is the initial metal concentration in the liquid phase, mg/L; C_e_ is the equilibrium Lu^3+^ concentration in the liquid phase, mg/L; V is the volume of solution, L; m is mass of specimen, g.

The sorption capacity of the solids (a_m_, mg/g) was estimated from experimental isotherms Q_e_ = f(C_e_).

The microspheres loaded with lutetium by contacting the concentrated solutions (1000 mg/L Lu^3+^) were thermally treated in air atmosphere by two different ways—in a fixed bed at 1273 K for 3 h in a muffle furnace and in a fast heating-cooling cycle of the moving bed at 1473 K for 3–5 sec using a vertical tube furnace. The products thus prepared are denoted as xLu/NaX-1000, xLu/NaP1-1000 and xLu/ NaX-1200, xLu/NaP1-1200, accordingly (x—Lu^3+^ concentration in solution).

### 2.4. Characterization Techniques

Powder X-ray diffraction (PXRD) data were collected on DRON-3 (IC “Bourevestnic”, St. Petersburg, Russia) diffractometer using Cu Kα radiation (2θ range 5–70°). The samples were prepared by grinding with octane in an agate mortar and packed into a flat sample holder for the PXRD measurements in the Bragg–Brentano geometry. The crystallographic database of the Joint Committee on Powder Diffraction-International Centre for Diffraction Data (JCPDS-ICDD, now known as ICDD) JCPDS-ICDD PDF-2 Release 2004 and software PhasanX 2.0 (V. 2.0, XraySite.com) were used to process PXRD patterns.

Morphologies and elemental composition of product particles were studied by scanning electron microscopy (SEM) using TM-3000 and TM-4000 (Hitachi, Tokyo, Japan) instruments equipped with the Bruker microanalysis system (Bruker, Berlin, Germany) including an energy-dispersive X-ray spectrometer with a XFlash 430 H detector and QUANTAX 70 software (V. 1.2, Bruker Nano GmbH, Berlin, Germany). The Pt layer with a thickness of 1–1.2 nm was deposited on the specimen surface to ensure a good electron current and improve the quality of shooting.

Simultaneous thermal analysis (STA) was performed on a TG-DSC STA 449C analyzer equipped with an Aeolos QMS 403C mass spectrometer (NETZSCH, Selb, Germany). The measurements were carried out under dynamic 20 % O_2_-Ar atmosphere at ambient pressure on heating in the range of 313–1373 K in Pt crucibles with perforated lids (a sample mass 10–12 mg; β = 10°/min). Qualitative composition of gas phase was determined by online QMS in the Multiple Ion Detection mode from the intensity of ions *m*/*z* = 18 (H_2_O), 32 (O_2_), and 44 (CO_2_).

The specific surface area (SSA, m^2^/g) was determined by the Brunauer–Emmett–Teller (BET) method [28] based on N_2_ adsorption isotherm measurements at 77 K using the Nova 3200e analyzer (Quantachrome Instruments, USA) and NovaWin Software.

The Lu^3+^ concentration in the solutions before and after sorption was measured by inductively coupled plasma mass spectrometry (ICP-MS) (XSeries II, Thermo Scientific, Waltham, MA, USA).

## 3. Results and Discussion

### 3.1. Hydrothermal Synthesis of Zeolitized Microspheres

Figure 2 shows PXRD patterns of the product particles formed in the NaOH-H_2_O-(SiO_2_-Al_2_O_3_)_glass_ system at 353 K and 393 K, accordingly (Table 1). Microstructures and elemental compositions of the materials are given in Figure 3 and Figure 4. The specific surface areas of the NaX and NaP1 designated products were 204 and 33 m^2^/g, respectively.

As expected, the monozeolite phases of cubic NaX (FAU, ICDD 00-012-246) (Figure 2a) and hexagonal NaP1 (GIS, ICDD 00-016-0354) (Figure 2b) were identified in the products. By the SEM data, the product particles preserve the spherical form of the glass precursor (Figure 3a and Figure 4a) and look like hollow microspheres with polycrystalline coverings. In the case of the NaX-bearing microspheres, the zeolite crystals have habits that are characteristic of faujasite-type zeolites, such as truncated octahedra, multi-faceted spherulites with 111 faces exposed [26]. The 3–5 μm crystals are supported by the unreacted glass (G region in Figure 3b). Applying other hydrothermal conditions (No 2 in Table 1), a compact layer of randomly oriented submicrosized columnar crystals form on the glass surface (Figure 4a,b). According to EDX measurements, the silica-alumina compositions of both zeolite phases in terms of the Si/Al ratio are in variation limits reported for zeolites NaX and NaP [23]. At the same time, if the Na/Al ratio in faujasite is close to 1 (Figure 3c-1), the same value for NaP1 does not exceed 0.7 (Figure 3c-1). It can be assumed that, since the structure of gismondin is able to include calcium and potassium [23], these cations can transfer from the cenosphere’s glass into zeolite structure upon its formation and compensate the negative charge of a zeolite framework jointly with sodium. 

### 3.2. Lu^3+^ Sorption and Solid-Phase Transformation of the Lu^3+^-Loaded Microspheres

The data of experimental measurements of Lu^3+^ sorption on the zeolitized microspheres are presented in Figure 5. As shown, the NaX-bearing microspheres are characterized by two times higher lutetium loading compared to the NaP1 microspheres, the average maximal sorption capacities (a_m_) for the microspheres being about 130 and 70 mg/g Lu^3+^, respectively. The higher Lu uptake of NaX can arise from the fact that this zeolite has a higher pore aperture than NaP1, which is a small pore one with a much less specific surface area [23]. It should be notified that the calculation of the capacity values does not take into account the degree of glass phase transformation and refer to all matter of the microspheres consisting of crystalline (zeolite, quartz, mullite) and glass phases. 

TG-DSC study was carried out to estimate the temperature of solid-phase transformation of Lu^3+^ loaded material. Figure 6 shows the thermal behavior of the Lu^3+^ free and Lu^3+^ loaded (Q_e_~140 mg/g Lu^3+^) NaX microspheres.

As shown, the heating of the samples in the range of 313–1373 K was accompanied by a continuous mass loss up to temperatures of up to 773 K. By the MS analysis of off-gases, this loss is caused by the zeolite water evolving occurred with maximal rates at 412 and 400 K for NaX and 140Lu/NaX, accordingly. The exothermic peaks at 1141 K and 1170 K can be assigned to the solid-state faujasite-to-nepheline transformation of neat NaX [22], and, in the case of Lu^3+^-loaded NaX microspheres, to recrystallization of a faujasite framework with participation of sorbed Lu^3+^. 

Based on the TG-DSC data, the thermal treatment of the Lu^3+^-loaded NaX and NaP1 microspheres was carried out in a fixed bed at 1273 K for 3 h and in a moving bed at 1473 K for 3–5 sec contact time with the high-temperature zone. As revealed by PXRD data, the long-term calcination at 1273 K resulted in the crystallization of monoclinic lutetium pyrosilicate Lu_2_Si_2_O_7_ (ICDD 00-034-509) in both zeolite systems (Figure 2a,b), this phase being a major component of crystalline constituents of the calcined microspheres. Nepheline (ICDD 01-079-0992), quartz (ICDD 83-2187), and mullite (ICDD 01-079-1454) are also identified as minor crystal phases. 

By SEM data (Figure 7a,b), the Lu_2_Si_2_O_7_ phase crystallizes as needle-like crystals that are incorporated in the bulk of former faujasite crystallites attached at both external and internal surfaces of microspheres. According to EDX examination, lutetium is unevenly distributed over the microsphere surface with a maximal content of up to 10 at. % Lu in the sites of faujasite localization. 

As for the Lu^3+^ containing NaP1 microspheres, there are no pronounced local inhomogeneities of lutetium concentration over the zeolite-derived surface layer after calcination at 1273 K (Figure 8). The almost smooth surface without growths and cavities, uniform Lu distribution, and Lu content of up to 16 at. % are distinctive features of the Lu encapsulated microspheres based on the NaP1 precursor.

The thermal treatment of the Lu^3+^-loaded zeolite precursors in another mode—fast heating–cooling by passing through the vertical tube furnace at 1473 K—resulted in (i) softening the glass-crystalline matter of the NaX bearing precursor with preserving its spherical form and inhomogeneous distribution of lutetium as well as (ii) partial elimination of surface open pores (Figure 9). 

In turn, the globule microstructure of the NaP1 bearing precursor and a character of Lu distribution has not changed significantly after heating at 1473 K (Figure 10). The surface of the NaP1 based microspheres has a granular relief (Figure 10c,e) and resembles the morphology of the untreated NaP1 covered surface (Figure 4b,c). This points to the fact that applied temperature is insufficient for softening the aluminosilicate material with the greater Lu content compared to the NaX based precursor. For both precursors, amorphization of the zeolite phases has taken place and the Lu_2_Si_2_O_7_ phase has not crystallized at 1473 K (Figure 2a,b). 

Thus, the short-time heat treatment of the Lu^3+^-loaded zeolitized microspheres at a softening temperature can provide the Lu^3+^-encapsulated glassy microspheres with the near smooth surface and homogeneous distribution of lutetium in a solid phase. By long-time heating the Lu^3+^-loaded zeolite-based precursors at temperature of solid-phase transformation, lutetium is accommodated in the monoclinic Lu_2_Si_2_O_7_ phase. The latter option preserves the crystal-like cover of the microsphere surface and results in immobilizing lutetium in a chemically stable crystalline form. 

## 4. Conclusions

For the first time, the cenosphere-based methodology to fabricate microsphere precursors of radiolabeled microspheres for application in brachytherapy was tested. The proposed concept includes the formation of zeolite on the surface, Lu^3+^ sorption by zeolite (to provide the sufficient Lu quantity) with subsequent solid-phase transformation under heating (to provide the formation of insoluble Lu compounds). The reactivity of the cenosphere’s aluminosilicate glass with respect to conversion into zeolite structures in an alkaline medium was used to elaborate microporosity in a cenosphere shell and to create binding sites for Lu^3+^ cations by means of ion exchange 3Na^+^ ↔ Lu^3+^. Formation of zeolitized coverings based on two types of zeolite structures, such as NaX (FAU) and NaP1 (GIS) that are differed by the Lu^3+^ sorption capacity, crystal size and morphology, compactness of the arrangement, and thermal behavior under heating, was successfully implemented. As revealed, the gismondin-based precursor provides the uniform Lu distribution and acceptable Lu content (up to 15 mol.% Lu_2_O_3_) in the solid phase and, due to this, can be considered as a promising candidate for further detailed study its applicability as a precursor of beta-emitting microspheres. The different regimes of heat treatment of Lu^3+^-loaded microspheres have resulted in two different forms of Lu encapsulation in the aluminosilicate matter, such as (i) the crystalline Lu_2_Si_2_O_7_ phase and (ii) the amorphous glass. The adjustment of the heating procedure with the simultaneous testing of the lutetium solubility in blood-like liquids is required at the next stages of the study.

## Figures and Tables

**Figure 1 materials-15-07025-f001:**
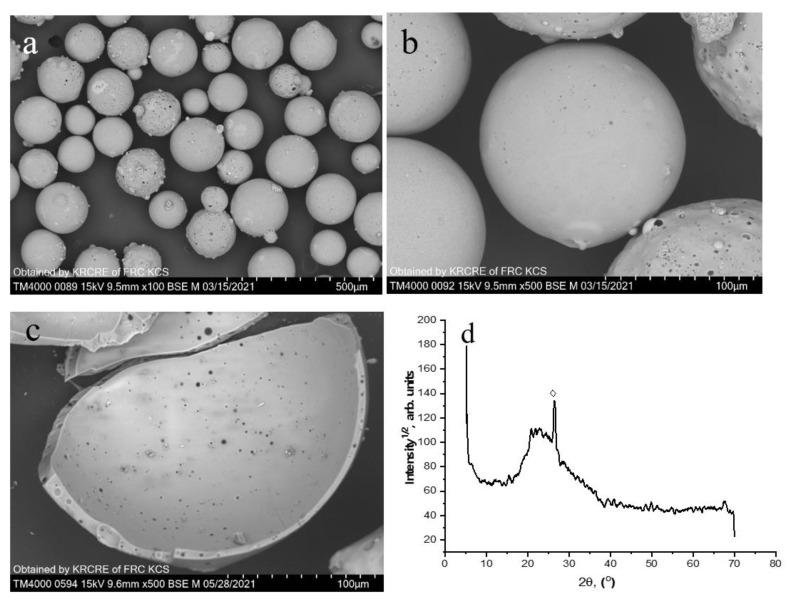
SEM images of initial cenospheres (**a**,**b**) and broken cenosphere globule (**c**); PXRD pattern of the cenospheres (**d**) (◊—quartz).

**Figure 2 materials-15-07025-f002:**
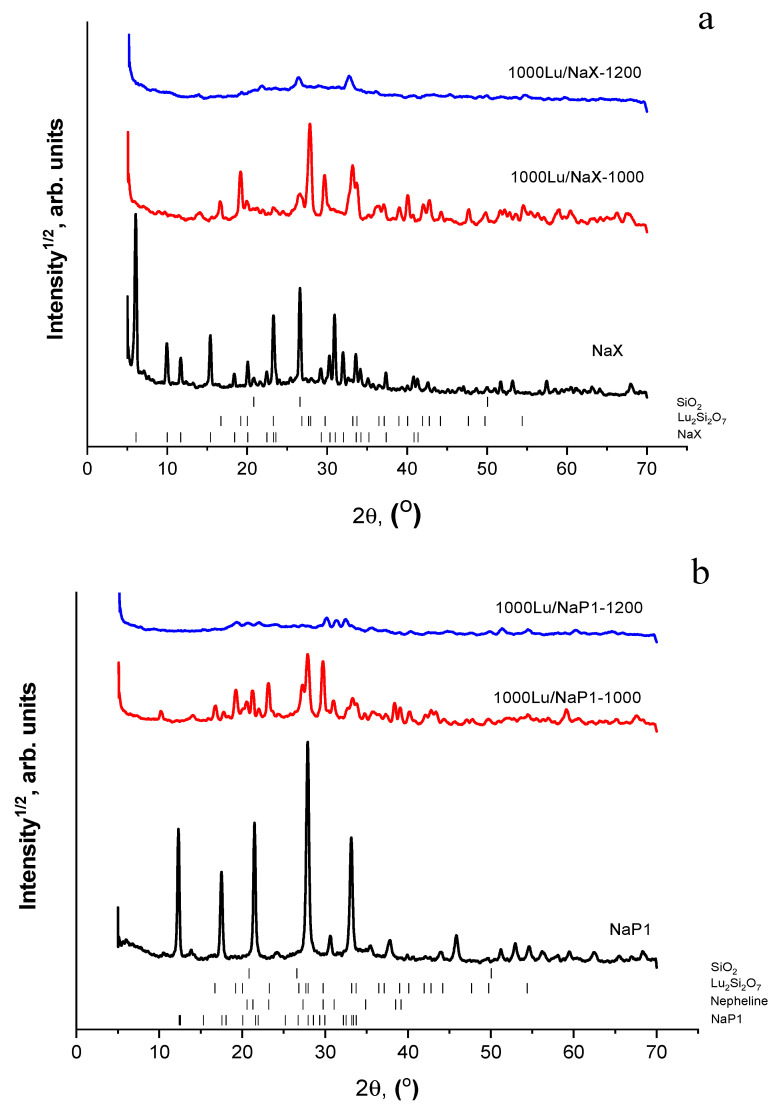
PXRD patterns of products resulted from hydrothermal processing of the NaOH-H_2_O-(SiO_2_-Al_2_O_3_)_glass_ system at (**a**) 353 K and (**b**) 393 K; products of solid-phase recrystallization of the Lu^3+^ loaded NaX (**a**) and NaP1 (**b**) bearing microspheres at 1273 K (1000Lu/NaX-1000, 1000Lu/NaP1-1000) and 1473 K (1000Lu/NaX-1200, 1000Lu/NaP1-1200).

**Figure 3 materials-15-07025-f003:**
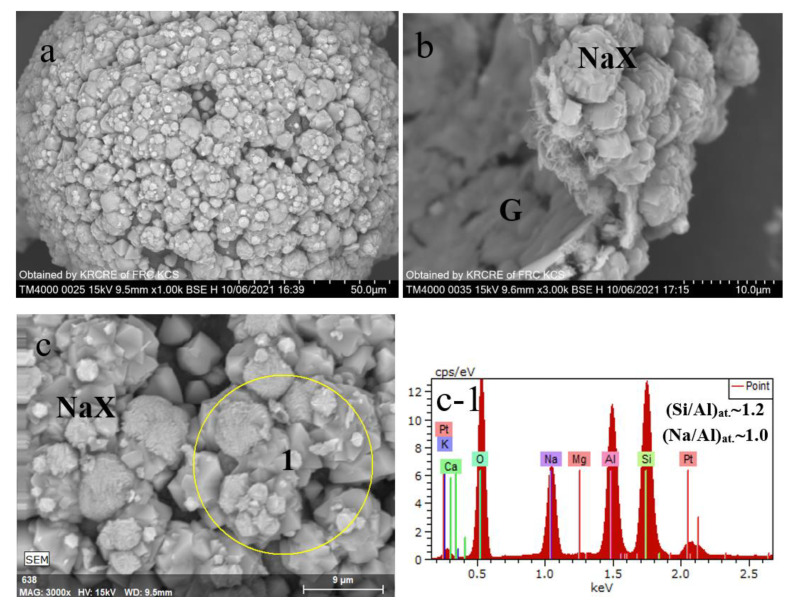
SEM images of product particles resulted from hydrothermal crystallization of cenosphere glass at 353 K (**a**–**c**); EDX spectra (**c-1**) for the local part of the zeolitized layer (**c**).

**Figure 4 materials-15-07025-f004:**
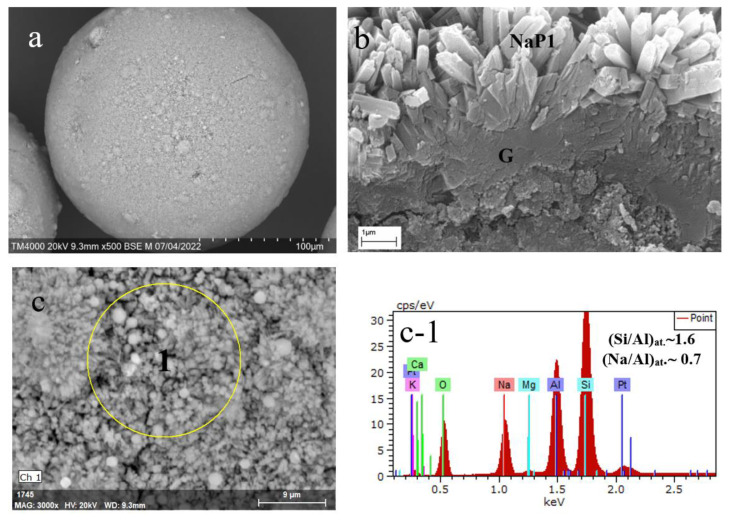
SEM images of product particles resulted from hydrothermal crystallization of cenosphere glass at 393 K (**a**), cross-section of the zeolitized shell (**b**); EDX spectra (**c-1**) for the local part of the zeolitized layer (**c**).

**Figure 5 materials-15-07025-f005:**
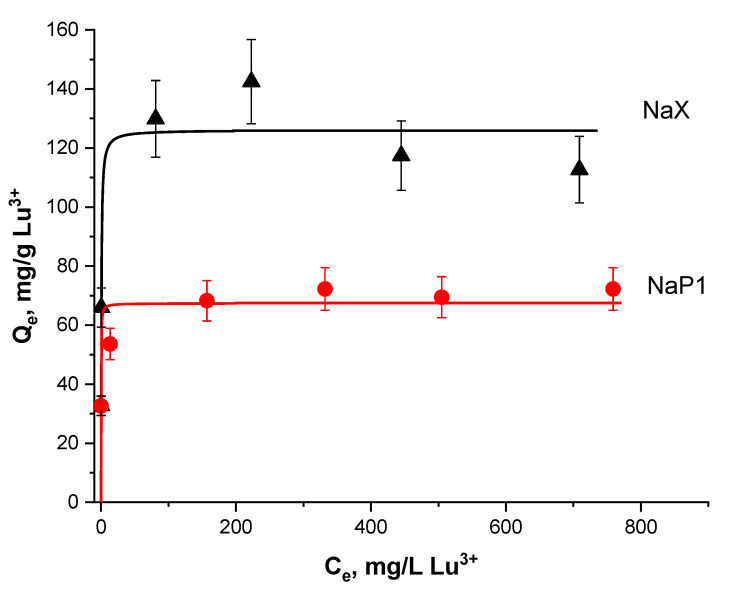
The Lu^3+^ sorption on the NaX and NaP1 bearing microspheres at room temperature: points—experimental data; lines—guide for eyes.

**Figure 6 materials-15-07025-f006:**
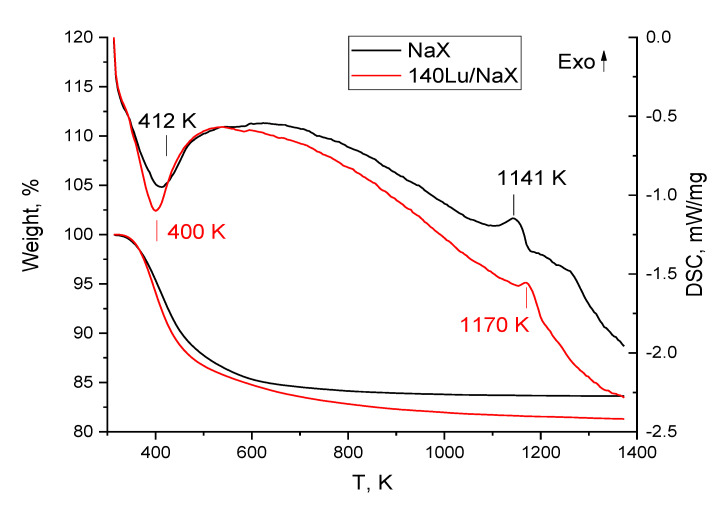
TG-DSC curves for the NaX microspheres without Lu^3+^ (NaX) and loaded with Lu^3+^ (140 Lu/NaX).

**Figure 7 materials-15-07025-f007:**
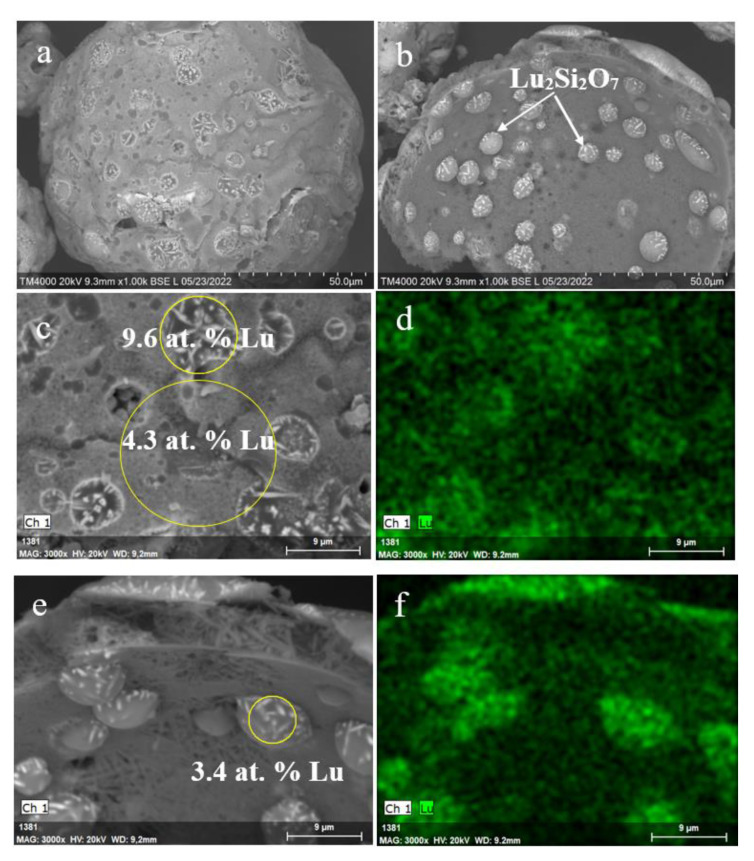
SEM images of the Lu^3+^ loaded NaX microspheres (**a**), internal surface of the microsphere globule (**b**), and local parts of the microsphere external (**c**) and internal surface (**e**) after thermal treatment at 1273 K; Lu distribution maps (**d**,**f**) for the local parts (**c**,**e**) of the microspheres.

**Figure 8 materials-15-07025-f008:**
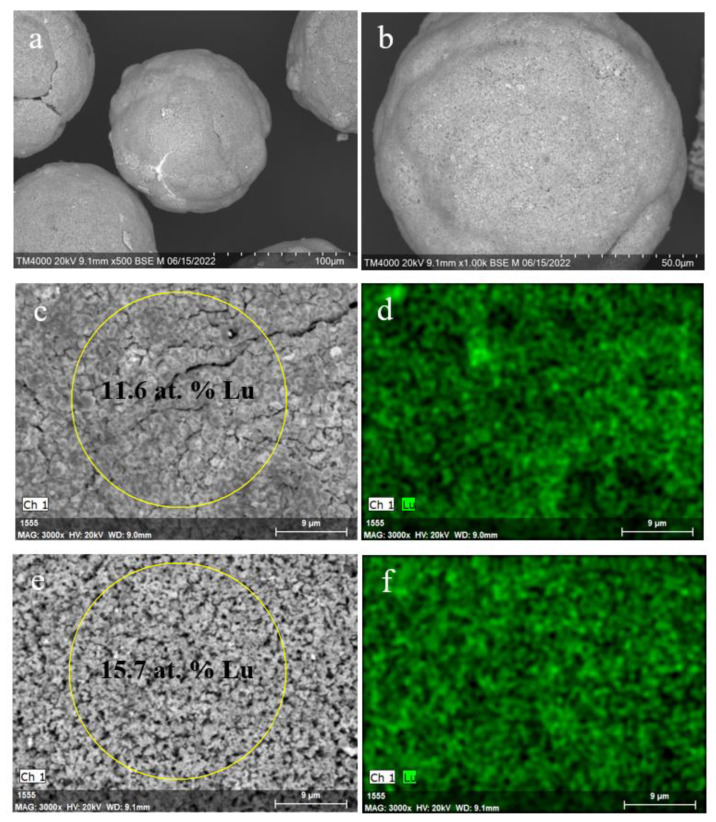
SEM images of the Lu^3+^ loaded NaP1 microspheres (**a**,**b**) and local parts of the microsphere external surface (**c**,**e**) after thermal treatment at 1273 K; Lu distribution maps (**d**,**f**) for the local parts (**c**,**e**) of the microspheres.

**Figure 9 materials-15-07025-f009:**
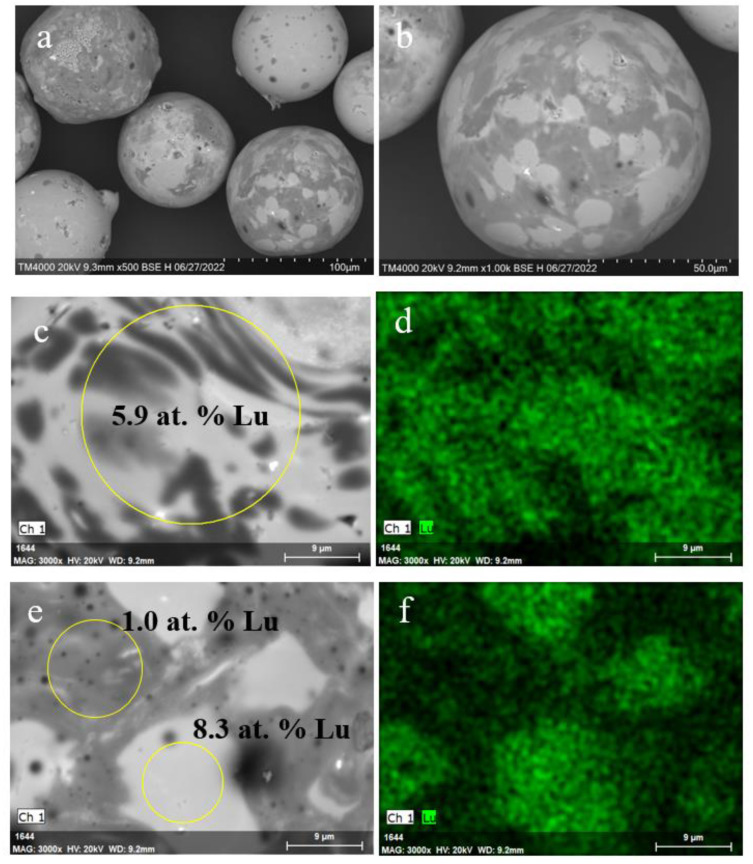
SEM images of the Lu^3+^ loaded NaX microspheres (**a**,**b**) and local parts of the microsphere external surface (**c**,**e**) after thermal treatment at 1473 K; Lu distribution maps (**d**,**f**) for the local parts (**c**,**e**) of the microspheres.

**Figure 10 materials-15-07025-f010:**
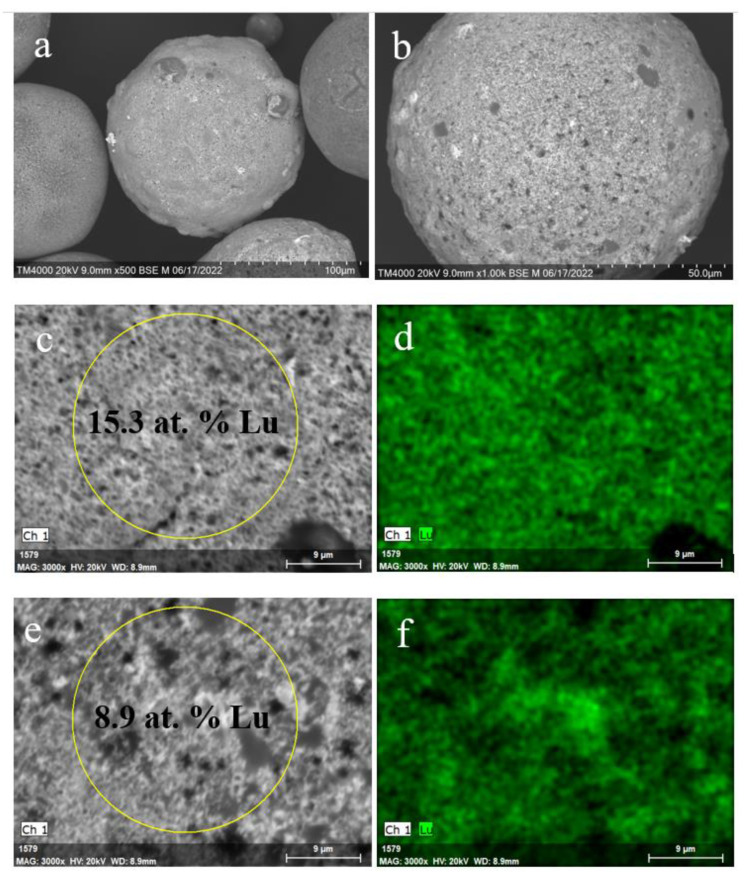
SEM images of the Lu^3+^ loaded NaP1 microspheres (**a**,**b**) and local parts of the microsphere external surface (**c**,**e**) after thermal treatment at 1473 K; Lu distribution maps (**d**,**f**) for the local parts (**c**,**e**) of the microspheres (**d**,**f**).

**Table 1 materials-15-07025-t001:** Molar compositions of the NaOH-H_2_O-(SiO_2_-Al_2_O_3_)_glass_ systems and parameters of syntheses.

No.	System	Sample	T, K	NaOH, mol/L	τ, h
1	1.0 SiO_2_/0.18 Al_2_O_3_/2.2 Na_2_O/100 H_2_O	NaX	80	2.5	48
2	1.0 SiO_2_/0.18 Al_2_O_3_/1.3 Na_2_O/100 H_2_O	NaP1	120	1.5	24

## Data Availability

Data sharing not applicable.

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
