# Peer review of "Cenosphere-Based Zeolite Precursors of Lutetium Encapsulated Aluminosilicate Microspheres for Application in Brachytherapy"

_materials, 2022, doi:10.3390/ma15197025_

Round 1
Reviewer 1 Report
Vereshchagina et al. reported the synthesis of cenosphere-based zeolite precursors for lutetium encapsulation aiming for further application in brachytherapy. The paper is well-written and good-looking. However, I detected several issues related to the zeolitic part of the concept brought by the authors. Below you will find my comments and questions point-by-point:
- What is the relevance of obtaining a zeolite from cenosphere-based microspheres for further zeolite destruction by thermal treatment at such high temperatures? Is the treatment at high temperatures important for brachytherapy applications? It seems the authors are preparing materials and then destroying them. Or the zeolites are serving as mere ion exchangers for Lu capture? This point must be revisited in the introduction and discussion part of the text before a further decision on the acceptance of the manuscript.
- In line 105, what does -180+80 µm means?
- Since authors attribute phase compositions from XRD, the peaks must be indexed.
- “M” is no longer used. The term must be replaced by mol/L.
- “°C” seems to be the unit chosen for the study, so “K” must be substituted.
- The first part of the Results and Discussion section could be part of the introduction of the paper.
- Numbers in the Y axis must be removed from XRD images and replaced for scales since X-ray diffractometers are starting from different Y values. Moreover, “Ne” should be properly explained in the legend in Figure 2.
- A table for quantitative results derived from EDX would be a benefit for readers.
- A better explanation for higher Lu uptake of NaX resides in the fact that this zeolite has higher SBET and pore aperture than NaP1, which is a small-pore one with much less SBET.
- The number of papers from the same group in the references is quite high, I suggest decreasing it.
After the raised above, I suggest major revisions for the paper to meet the standards of Materials journal.
Reviewer 2 Report
| The authors demonstrated the cenosphere-based methodology to fabricate microsphere precursors of radiolabeled microspheres for application in brachytherapy,the method is pretty creative and important, this manuscript can be accepted for publishing. | |
Author Response
No comments and Suggestions for Authors
Reviewer 3 Report
In this paper, the cenosphere-based methodology to fabricate microsphere precursors of radiolabeled microspheres for application in brachytherapy was tested. It is an interesting content, but arranged structure needs to be further improved. Therefore, it needs minor revision before it is published in this journal. Please check the attachment.

Round 2
Reviewer 1 Report
The authors answered my point-by-point questions and the paper can be accepted in its present form.